# Perceived Barriers to Implementing Education for Sustainable Development among Korean Teachers

**Woonsun Kang**

Department of Social Studies Education, Daegu University, Gyeongsan-si, Gyeongsangbuk-do 38453, Korea;
wskang@daegu.ac.kr; Tel.: +82-053-850-4144

**Abstract:** The objective of this study was to identify whether there are any homogeneous subclasses of teachers exhibiting different profiles of barriers to implementing ESD among Korean secondary teachers, and to examine whether teachers' experiences of taking an ESD course in their pre-service teacher education and in their in-service training on ESD are predictive of membership in subclasses of perceived barriers to implementing ESD. Korean secondary teachers from various subjects were selected as a sample. I carried out latent class analysis (LCA) on barrier variables and assessed the association of both the experiences of taking an ESD course in their pre-service teacher education and in-service training on ESD with membership in the latent subclasses using multi-nominal logistic regression. These analyses were performed using PROC LCA. Research results are as follows: Firstly, four latent classes were identified: the few barrier, the individual barrier, the combination of individual and class-driven structural barrier, and the combination of individual and structural barrier. Secondly, both the experiences of taking an ESD course in their pre-service teacher education and in-service training on ESD were significant predictors of latent class membership. The current study could potentially assist both pre-service teacher educational institutions and in-service teacher training organizations with strategies designed to improve ESD competency among teachers.

**Keywords:** education for sustainable development; perceived barriers; the individual barrier; the structural barrier

## 1. Introduction

To create a more sustainable world, individuals are equipped with the requisite knowledge, skills, values, and attitudes that empower them to contribute to sustainable development (SD). Education for Sustainable Development (ESD) is education that allows every human being to acquire the knowledge, skills, attitudes, and values necessary to shape a sustainable future. People cannot acquire the knowledge, skills, and values required to achieve a sustainable society without education. Therefore, ESD has been recognized as a key instrument to achieve the SD, and has been emphasized as an important tool in increasing public awareness and understanding of SD [1].

By 2030, ensure that all learners acquire knowledge and skills needed to promote SD, including, among others, through education for sustainable development and sustainable lifestyles, human rights, gender equality, promotion of a culture of peace and nonviolence, global citizenship and appreciation of cultural diversity and of culture's contribution to sustainable development [2].

Korea has developed a national strategy which includes strong emphasis upon the value of ESD. Sustainable development is not included as a separate subject in the Korean school system. However, SD is addressed in the subject syllabuses such as Ethics, Social Studies, and Science. Moreover, SD is selected as one of ten cross-curriculum priorities to be incorporated into the Korean Curriculum, and policy-makers have encouraged teachers to integrate SD within the various existing school subjects as cross-curriculum. All teachers are, therefore, expected to include SD in their teaching.

But 40% of Korean teachers surveyed do not implement ESD [3], which imply that these efforts do not guarantee the successful implementation of ESD in schools, and a critical point in the successful implementation of ESD strongly depends on teachers. So why do Korean teachers not carry out ESD?

One of the answers to this question could be found in teachers' perceived barriers. This is because teachers are less likely to integrate SD into existing school subjects, when teachers perceived barriers. In other words, teachers will be more likely to implement ESD in their practice if perceived barriers are low. Gaining an understanding of perceived barriers is crucial to facilitating ESD. This understanding will increase the likelihood of successful ESD implementation in the practice.

Thus, identifying perceived barriers to ESD implementation is a good way to improve the possibilities of successful ESD implementation. Prior researchers [4–9] have reported that a lack of the knowledge about SD, lack of interest, lack of instructional materials, over-crowded curriculum, and lack of time as relatively common barriers for teachers. Most of them focused on frequency distributions and simple cross tabulation of responses using individual scales in isolation. This approach has helped us explore the relative abundance of each particular target data within their sample. In spite of this significance, there is a limitation that is placed on such research methods. These methods do not tell us whether there are certain combinations that generally co-occur. But, perceived barriers to ESD implementation may co-occur and interact in ways that differentially impact teachers' implementation. To gain a more thorough understanding of the perceived barriers, it is necessary to reveals co-occurring barriers that teachers may simultaneously perceive, not to approach perceived barriers simply as single and independent variables.

One alternative method that can assess co-occurring barriers is latent class analysis (LCA), which is an analysis method for uncovering hidden patterns of association that can exist between observations. This method is a useful tool for identifying a small set of underlying subgroups of individuals based on the intersection of multiple observed characteristics [10–14]. By reducing the number of subgroups a solution can more readily lend itself to drawing perceived barriers and promotion implications. Finally, identifying types of perceived barriers to ESD implementation is an effective approach to increase the possibilities of successful ESD implementation.

While identifying types of perceived barriers is a pivotal step for successful ESD implementation, it is equally crucial to explore factors that may reduce perceived barriers. Prior researchers [4–9] have reported that major perceived barriers to ESD implementation are lack of the knowledge on SD, lack of awareness on SD, and lack of pedagogical knowledge on SD. Most of them may be coping with an increase in competency. One of the ways for teachers to improve one's competency is to take teacher education courses. Pre-service teacher education and in-service training can play an important role in strengthening teachers' competency [15–20]. While teachers' experiences of taking ESD courses in their pre-service teacher education and in their in-service training on ESD may have differential effects on perceived barriers, researchers have not systematically explored whether teachers' experiences of taking ESD courses in their pre-service teacher education and in their in-service training on ESD has an influence or not.

Thus, I aim to identify the types of perceived barriers that prevent Korean teachers from implementing ESD, and to analyze if teachers' experiences of taking ESD courses in their pre-service teacher education and in their in-service training on ESD are predictive of membership in perceived barrier types. Against this backdrop, the following two research questions were addressed:

Research Question 1: Is there a latent class structure that represents the heterogeneity in the barriers to ESD implementation among Korean teachers? If so, what types of barriers to ESD implementation exist among Korean teachers?

Research Question 2: Are teachers' experiences of taking ESD course in their pre-service teacher education and in their in-service training on ESD predictive of membership in perceived barrier types?

## 2. Literature Review

Studies have explored the constraints faced by teachers in implementing ESD in schools and these have been identified as complex conceptualizations of SD, a lack of knowledge as it relates to ESD, the pressures of an over-crowded curriculum, lack of funding, and resources, etc.

First of all, the term SD is a complex and evolving concept, which causes a barrier in clarifying the term. The complexity of SD issues poses many challenges to teachers. It has been reported that there are some areas within the higher education sector where the concept of sustainability is not yet fully understood [21,22]. Spiropoulou et al. reported that teachers hold a misunderstanding of the conceptual meaning of sustainability [7]. Because SD is hard to define, it is also difficult to teach. A study reported that a highly complex and wide-ranging concept of SD is a barrier to ESD [23].

Without the necessary content knowledge (CK) and pedagogical knowledge (PK) related to ESD, we cannot expect to implement a meaningful teaching practice. The lack of necessary CK and PK related to ESD can be a barrier to ESD implementation. Several studies [9,15,24,25] reported the lack of necessary competency as a perceived barrier to implementing ESD.

Regarding the potential effect of beliefs on teachers' implementations, some studies were also conducted in order to reveal teachers' beliefs on ESD. The studies reported that teachers are unaware of the need for ESD [26,27]. Some previous studies [4,6,9,26–28] have identified teachers' beliefs and attitudes as barriers to ESD. Lack of awareness is major barriers to ESD implementation [29]. ESD is not taught as a separate subject. In this situation, ESD becomes an "add-on" to an already overcrowded curriculum, and can constitute a significant structural barrier [4,9,30,31]. University instructors may also see ESD as peripheral to their specialized subject matter [32].

ESD appears to rely on a common vision or objective among teachers and school leaders. The principal is considered to play a very important role in whether a school becomes engaged with ESD. Above all, principals have a powerful influence on class time schedules, and the allocation of resources for expenses associated with funding for teaching materials for ESD. Wu and Wang [33] highlight the importance of proactive leadership. It is critical to support the introduction of ESD in school through changes in curricula, vision or through more strategic intervention. Therefore, a disinterested principal can be a key barrier to ESD. The studies [4,9,25] pointed out that teachers perceive a lack of support from principals.

It has been reported that the lack of effective access to materials has also acted as a great barrier to implementing ESD [4,6,9]. Other barriers to implementing ESD are: the perceived irrelevance of ESD to some disciplines, lack of time in the curriculum [4,6,9], the lack of financial resources [4,9], and a lack of student motivation for ESD [4].

I compared the descriptive analysis results of perceived barriers to ESD. The major barriers of ESD faced by Korean elementary school teachers were the lack of teaching material (22.7%), too much administrative work and lack of time (19.2%), a lack of awareness on ESD (17.2%), and ESD not being included in one's teaching contents (9.9%) [6].

Swedish teachers have a positive attitude towards SD, as a total of 90% stated that they were passionate advocates or thought that it was a good thing. Despite this, more than half of those teachers (55%) felt that they had difficulties integrating SD into their teaching. The most common barriers reported were that they lacked inspiring examples of how to include SD issues into their teaching (29%), lacked necessary expertise on SD (27%), lacked time to implement necessary changes to their courses (15%), and lacked sufficient support from the school management (7%). Some teachers also felt that SD was not relevant to their subject (14%) [24].

The mean scores of Turkish elementary teachers' responses on commonly perceived barriers to ESD were lack of the knowledge about SD (4.67), lack of knowledge about teaching SD (4.78), lack of instructional materials (4.40), large class sizes (4.28), lack of principal support (4.23), and lack of funding (4.14) [6]. In a survey, 92% of Australian teachers surveyed think that SD is important, of value to students, and should be integrated into the curriculum. However, 80% of teachers are either unaware of ESD or lack a clear understanding of SD. Additionally, teachers pointed out a lack of

a definition for SD (64%), lack of information about how SD relates to various subjects (56%), and curriculum materials such as syllabuses (46%) [4].

## 3. Research Methodology

### 3.1. Data Collecting and Sampling Sample and Procedures

The data was collected between 13 June 2018 and 21 October 2018. The current study utilized a survey research method. The convenience sampling method was preferred as a sampling method in the present study. The measurement tool was converted to an online survey and then sent to teachers as mail. A total of 211 teachers from various teaching fields responded to the survey. The more detailed demographic information of the teachers is displayed in Table 1.

**Table 1.** Teacher demographics.

| Characteristics | | Respondents | |
|---|---|---|---|
| Gender | Male | 61 | 30.3 |
| | Female | 140 | 69.7 |
| Age | 20s | 33 | 16.1 |
| | 30s | 81 | 39.5 |
| | 40s | 48 | 23.4 |
| | 50s | 39 | 19.0 |
| | 60s | 4 | 2.0 |
| Year of teaching experience | 5 years or less | 62 | 31.2 |
| | 6~10 years | 38 | 19.1 |
| | 10~15 years | 33 | 16.6 |
| | 16~20 years | 26 | 13.1 |
| | More than 20 years | 40 | 20.1 |
| Highest educational level | Bachelor | 153 | 74.7 |
| | Master | 50 | 24.3 |
| | Doctor | 2 | 1.0 |
| Size of school | ~ 17 Class | 37 | 18.8 |
| | 18 ~ 35 Class | 132 | 67.0 |
| | 36 Class ~ | 28 | 14.2 |
| Subject taught | Ethics | 4 | 2 |
| | Korean language | 28 | 13.9 |
| | Mathematics | 23 | 11.4 |
| | English | 20 | 9.9 |
| | Social studies | 36 | 17.8 |
| | Science | 20 | 9.9 |
| | Music | 3 | 1.5 |
| | Arts | 3 | 1.5 |
| | Physical education | 8 | 4.0 |
| | Technology home Economics | 8 | 4.0 |
| | Others | 49 | 24.3 |
| School's location | Large city | 126 | 72.0 |
| | City | 46 | 26.3 |
| | Village or rural area | 3 | 1.7 |
| Pre-service teacher education curriculum on ESD | Non-experienced | 173 | 84.8 |
| | Experienced | 31 | 15.2 |
| In-service training on ESD | Non-participation | 144 | 70.9 |
| | Participation | 59 | 29.1 |

*3.2. Measurement Instrument*

The research instruments used in this study was a questionnaire. The questionnaire consisted of two parts. The first part consisted of a checklist to determine gender, age, years of teaching experience, highest educational level, subject matter taught, participation in activities on SD/ESD, size of school, and the school's location to identify the socio-demographic background of the respondents. The second part listed perceived barriers to ESD implementation. The survey was employed to investigate the perceived barriers to ESD implementation among Korean secondary teachers. The perceived barriers to ESD implementation scale consisted of fourteen different statements that teachers may perceive as barriers. Survey items used in this investigation were borrowed from a portion of the original questionnaire of prior research [4,6,9]. Survey items in this research consisted of dichotomous questions which could be answered either Yes or No. For each item, the teachers were asked whether the description applied to him/herself (Yes or No). These are shown in Table 2.

**Table 2.** Measurement items.

| Questionnaire Items | Scale | Reliability |
|---|---|---|
| Have you perceived any of the following barriers for implementation of ESD? Pease indicate "Yes" or "No" for each specified item. | Lack of pedagogical knowledge (PK) in ESD<br>Lack of knowledge about SD<br>Lack of information<br>Confusion about SD<br>Lack of instructional materials<br>Lack of interest<br>Lack of awareness<br>Lack of incentives<br>Lack of principal support<br>Class size too large<br>Too much administrative work<br>Curriculum too overcrowded to add ESD<br>Lack of funding for materials<br>Not included in one's teaching contents | Yes: 1<br>No: 2 | 0.88 |

Note: The reliability was assessed through the Kuder–Richardson reliability coefficients (KR-20)

Content validity of the research instruments was assessed through a review by a panel of experts. The three-member panel included faculty members and a teacher with a doctorate in education. As a result of the review, the survey questions were modified to allow respondents to indicate teachers' experiences of taking ESD courses in their pre-service teacher education. To establish the component structure of items, I carried out non-linear principal component analysis (NLPCA) with the program CATPCA from the Categories module in SPSS [34].

Traditionally, principal components analysis (PCA) assumes that all of the variables are scaled at the numeric level, and the relationships between variables are linear. Principal components analysis suffers from limitations due to the presence of assumptions. Otherwise, the main advantages of NLPCA are that it takes into account the categorical nature of variables and it does not rely on normality and linearity assumptions [35]. Therefore, NLPCA enabled me to examine if responses from several barriers differed with respect to the component structure. All the scale items were subjected to NLPCA followed by varimax rotation. As displayed in Table 3, NLPCA yielded two components.

**Table 3.** Rotated component loadings of the two-component CATPCA solution.

|  | Component | |
|---|---|---|
|  | **Individual Barriers** | **Structural Barriers** |
| Lack of PK in ESD | **0.76** | 0.18 |
| Lack of knowledge about SD | **0.74** | −0.11 |
| Lack of information | **0.63** | 0.14 |
| Confusion about SD | **0.63** | 0.07 |
| Lack of instructional materials | **0.59** | 0.05 |
| Lack of awareness | **0.58** | 0.29 |
| Lack of interest | **0.41** | 0.25 |
| Lack of incentives | 0.19 | **0.66** |
| Lack of principal support | 0.16 | **0.65** |
| Class size too large | 0.01 | **0.62** |
| Too much administrative work | 0.08 | **0.61** |
| Curriculum too overcrowded to add ESD | −0.06 | **0.56** |
| Lack of funding | 0.22 | **0.48** |
| Not included in one's teaching contents | 0.21 | **0.42** |
| Variance accounted for (VAF) | 2.98 | 2.56 |
| Cronbach's α | 0.75 | 0.71 |

Note: The strong correlation of a variable to a component appears in bold

The two components of the perceived barriers to ESD can be interpreted in terms of the individual barriers and structural barriers. In the component 1, lack of PK in ESD, lack of knowledge about SD, lack of information, and confusion about SD are barriers related to teachers' competency. Furthermore, lack of instructional materials can be approached through the lens of teacher competency. Teachers should be able to create teaching materials that are appropriate to standards or contents. In the case of a lack of competency, teachers are forced to rely on outside experts for teaching materials. Therefore, component1is referred to as an individual barrier.

Otherwise, in component 2, lack of incentives, lack of principle support, non-inclusion in one's teaching contents, too much administrative work, curriculum too overcrowded, class size too large, and lack of funding are involved in structural barriers. Too much administrative work, a too overcrowded curriculum and/or class size, and lack of funding are involved in obstacles in the classroom. Lack of incentives, lack of principle support, and non-inclusion in one's teaching contents are related to institutional practices or policies. Consequently, component 2 is not related to individual beliefs, preferences, or identities, but can be understood as aspects of the external environment that limit ESD. Therefore, component 2 is referred to as structural barriers.

The reliability and internal consistency of the measurements were assessed through the Kuder–Richardson reliability coefficients (KR-20), which was used to measure the internal consistency reliability of a variable with dichotomous data. The KR-20 was 0.88, which means that the reliability was relatively high.

### 3.3. Analysis Method

I conducted descriptive analysis and inferential analysis. The descriptive statistics were carried out to examine the distribution for single variables. The LCA was conducted in order to determine answers to the second research question. The LCA aims to categorize same individuals into latent classes, in which each latent class is considered as consisting of homogeneous individuals with regards to the manifest items, and the different latent classes are considered as representing the unobserved heterogeneity among individuals in these manifest items. When categorical items are used, the latent class model has the advantage of making no assumptions about the distributions of the indicators other than that of local independence [36].

For this reason, I conducted an LCA. The AIC (Akaike information criterion), BIC (Bayesian information criterion), and entropy were used for selection of the optimal number of latent classes. The smaller the AIC and BIC for a particular mode, the better the trade-off between fit and parsimony [36]. The quality of classification was measured by entropy. The entropy evaluates the quality of the measurement instrument as a whole. The entropy value is between 0 and 1. The higher the entropy is the clearer the latent class identification. Entropy values over 0.8 denote a good classification of the latent classes [37]. With technical value, each class should be discernible from the others on the basis of the item-response probabilities, and it should be possible to assign a meaningful label to each class and no class should be minor in size [36]. Multi-nominal logistic regression was carried out in order to explore whether teachers' experiences of taking ESD course in their pre-service teacher education and in their in-service training on ESD are predictive of membership in perceived barrier types.

Descriptive statistics, NLPCA, and KR-20 were performed using the Statistical Package for Social Sciences (SPSS Ver. 25.0). The LCA and multi-nominal logistic regression were carried out using PROC LCA, a SAS procedure for LCA developed for SAS Version 9.4 for Windows.

## 4. Results

### 4.1. What Barriers Do Korean Teachers Face in Implementing ESD?

I conducted a descriptive analysis to examine the distribution for single variables before running the LCA. Barriers to implementation of ESD consisted of fourteen different statements that teachers may face as barriers. Teachers' response percentages on barriers to implementing ESD can be seen in Figure 1.

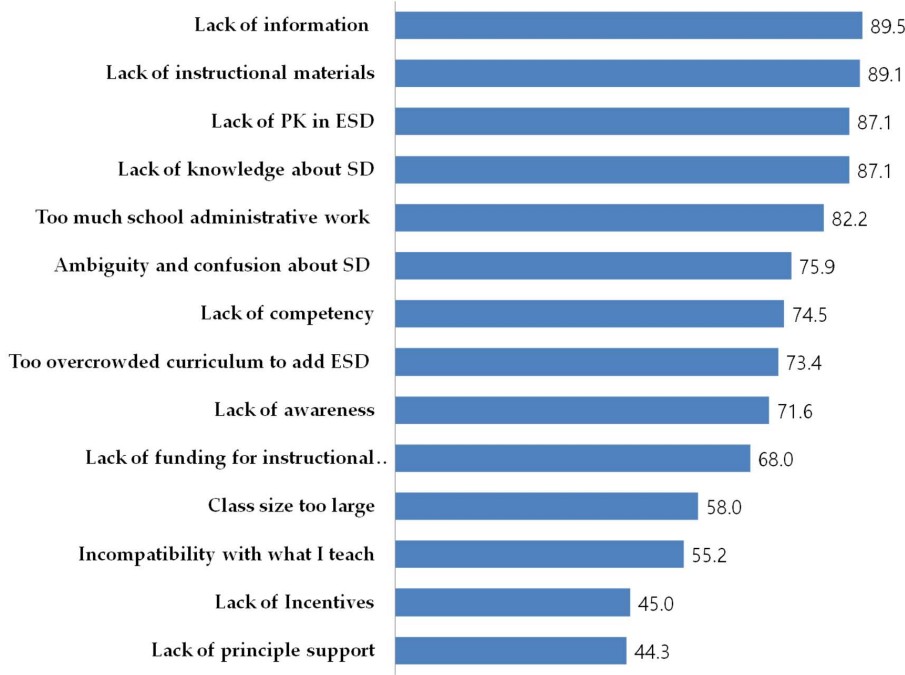

**Figure 1.** Percentage of teachers' responses on barriers to implementing ESD.

As can be seen in Figure 1, there were considerable perceived barriers to ESD implementation to the extent that: approximately 90% of Korean secondary teachers surveyed said they have a lack of information (89.5%), lack of instructional materials (89.1%), lack of knowledge about SD (87.1%), and lack of PK in ESD (87.1%). These perceived barriers were considered to exist with regard to comprehension and understanding of the concept and their teaching skills, and where teachers can access materials to help them implement ESD in their teaching practices, which belong to the individual barriers' component. Furthermore, there was considerable lack of interest (74.5%). Amongst Korean secondary teachers are confused about concept of SD (75.9%) and have lack of interest (74.5%).

For perceived structural barriers, Korean teachers responded as follows: too much school administrative work (82.2%), curriculum too overcrowded (73.4%), lack of funding (68.0%), class size too large (58.0%), not included in one's teaching contents (55.2%), lack of incentives (45.0%), and lack of principle support (44.3%).

Therefore, Korean teachers surveyed were more likely to perceive individual barriers than structural ones. Taken together, most teachers surveyed perceived many obstacles in the classroom: insufficient competency, lack of support or resources, large class sizes, and time constraints.

### 4.2. What Types of Perceived Barriers Exist among Korean Teachers?

To identify if there was a latent class structure that adequately represented the heterogeneity in perceived barriers to ESD among Korean teachers, and if so, what types were they, I carried out an LCA. Since two components were identified (Table 3), the underlying assumption in each was guided either by individual barriers, structural barriers, or the combination of the two, therefore, generating four possibilities. I repeated the use of PROC LCA for 2–4 class models each with 14 barrier indicators, and the point was to explore possible baseline models. Model fit statistics for LCA models with different numbers of latent classes are shown in Table 4.

**Table 4.** Comparison of baseline models.

| Model | AIC | BIC | Entropy |
|---|---|---|---|
| 2 class | 991.87 | 1087.81 | 0.78 |
| 3 class | 927.03 | 1072.60 | 0.80 |
| **4 class** | **893.88** | **1089.07** | **0.82** |

Note: Boldface type indicates the selected model. AIC: Akaike's information criterion; BIC: Bayesian information criterion.

As can be seen in Table 4, while the AIC values were the lowest, BICs were slightly larger in the four-class model compared to other classes; however, taken together with other statistics, the four-class basic model was considered as optimal. This is because the four-class basic model had the highest entropy of 0.82 with a better interpretation compared to others, also indicating a good quality of latent class classification.

In addition to the technical analysis, I explored whether the four-class model was interpretable. Specifically, I explored whether each class could be distinguishable from the others on the basis of the item-response probabilities, no class was trivial in size, and it was possible to assign a meaningful label to each class [36].

As shown in Table 5, the four-class model was distinguishable and non-trivial, and meaningful labels could be assigned to each class (Table 5.) Therefore, the four-class model was selected as the baseline model of barriers to implementing ESD among Korean secondary school teachers.

**Table 5.** Item response probabilities and probability of class membership for the four-class model.

| Component | Item Label | Latent Class | | | |
|---|---|---|---|---|---|
| | | Few Barrier | Individual Barrier | Combination of Individual and Class-Driven Structural Barrier | Combination of Individual and Structural Barrier |
| | 1. Lack of PK in ESD | 0.14 | 0.95 | 0.89 | 1.00 |
| | 2. Lack of knowledge about SD | 0.35 | 1.00 | 0.84 | 0.97 |
| Individual | 3. Lack of information | 0.44 | 0.89 | 0.93 | 0.98 |
| | 4. Confusion about SD | 0.16 | 0.91 | 0.57 | 0.98 |
| | 5. Lack of instructional materials | 0.48 | 1.00 | 0.89 | 0.95 |
| | 6. Lack of interest | 0.00 | 0.66 | 0.84 | 0.88 |
| | 7. Lack of awareness | 0.38 | 0.66 | 0.55 | 0.93 |

**Table 5.** *Cont.*

| Component | Item Label | Latent Class | | | |
|---|---|---|---|---|---|
| | | Few Barrier | Individual Barrier | Combination of Individual and Class-Driven Structural Barrier | Combination of Individual and Structural Barrier |
| Structural | 8. Lack of incentives | 0.15 | 0.00 | 0.31 | **0.80** |
| | 9. Lack of principal support | 0.05 | 0.00 | 0.46 | **0.70** |
| | 10. Too large class size | 0.34 | 0.21 | 0.62 | **0.76** |
| | 11. Too much administrative work | 0.47 | 0.44 | 0.81 | **1.00** |
| | 12.Too overcrowded curriculum | 0.43 | 0.25 | **0.90** | 0.84 |
| | 13. Lack of funding for materials | 0.24 | 0.35 | 0.75 | **0.86** |
| | 14. Not included in one's teaching contents | 0.29 | 0.42 | 0.21 | **0.90** |
| Probability of class membership | | 10% | 17% | 29% | 43% |

Note: Item response probabilities means the probability of answering "yes" to each question. The greater probabilities appear in bold font to highlight the overall pattern.

Each column in Table 5 shows the item-response probabilities for endorsing each item, probability of membership, and the assigned label for each class. Each row represents a different item, and the four columns of numbers are the item-response probabilities of answering "yes" to the item, given that someone belonged to that class. The item response probabilities indicate the probability for an individual to provide a certain response to a specific item given that she or he has been classified in a specific latent class, which show the differences in response patterns that help me distinguish the classes. The probabilities of class membership show the probability for an individual's membership in a certain class.

Looking at the pattern of responses for all the classes, I obtained an overall picture of the meaning of the four classes, which helped me label them appropriately and meaningfully. Generally, teachers in Class 1 were much less likely to respond "yes" to all variables than the other classes, while teachers in Class 4 were much more likely to respond "yes" to all variables than the other classes.

Those in Class 2 (95%), Class 3 (89%), and Class 4 (100%) had a high probability of saying "yes, lack of PK in ESD", while only 14% of those in Class 1 respond yes to lack of PK in ESD. In Item 8, 80% of those in Class 4 say they have a lack of incentives. By contrast, those in Class 1 (15%), Class 2 (0%), and Class 3 (31%) had a low probability of saying "yes, lack of incentives" For Item 13, few of those in Class 1 (24%) and Class 2 (42%) have a lack of funding for materials, while a considerable number of those in Class 3 (75%) and Class 4 (86%) said they have a lack of funding for materials.

As shown in Table 5, Class 3 was similar to Class 4 on nine items (1, 2, 3, 5, 6, 10, 11, 12, and 13). But for Item 8, 80% of those in Class 4 said they have a lack of incentive, while only 30% of Class 3 said they have a lack of incentive. In addition, for Item 14, 90% of those in Class 4 said that ESD was not included in one's teaching contents, while only 20% of Class 3 said that ESD was not included in one's teaching contents. Focusing just on each class, the characteristics of each class are as follows:

Class 1 (the Few barrier): Less than 50% of teachers in Class 1 were likely to respond "yes" to all variables. Few of them had a lack of knowledge about SD (35%), few had a lack of PK in ESD (14%), and few were confused regarding SD (16%). Especially, for lack of competency (0%) and lack of principle support (5%), they rarely answered "yes". It seems that those in Class 1 reported very few barriers. Therefore, this class was named the "few barrier" type. Only 10% of teachers belonged to the few barrier type.

Class 2 (the Individual barrier): Those in Class 2 had a high probability of saying "yes" to the individual component, while they had a low probability of saying "yes" to structural components. For Item 2 and 5, the probability of answering "yes" to each question was 100%. By contrast, the probability

of answering "yes" to Items 10 and 12 was only 20%. Furthermore, for Items 8 and 9, the probability of answering "yes" to each question was 0%. Therefore, this class was labeled the "individual barrier" type. Approximately 20% of teachers belonged to this class

Class 3 (the Combination of Individual and Class-Driven Structural barrier): Those in Class 3 had a high probability of saying "yes" to the individual components. Those in Class 3 had a high probability of saying "yes" to some of the structural barriers (10, 11, 12, and 13), while they had a low probability of saying "yes" to other structural barriers (8, 9, and 14). Class size, administrative work, and too overcrowded curriculum are related to class circumstances. A lack of funding for materials can be directly related to class-driven barriers of structural barriers. On the other hand, non-inclusion in one's teaching contents is related to the subject's syllabus in compulsory schools, and incentives and principal support can be relevant to institutional practices and policies. As a result, lack of incentives, lack of principle support, and non-inclusion in one's teaching contents can be related to institutional barriers. Based on these characteristics, this class was termed the "combination of individual and class-driven structural" barrier type. Almost 30% of total respondents belonged to this type.

Class 4 (the Combination of Individual and Structural barrier): Those in Class 4 have lack of PK in ESD (100%), lack of knowledge about SD (97%), lack of information (98%), lack of instructional materials (95%), lack of awareness (93%), lack of interest (88%), lack of funding for materials (86%), and lack of incentives (80%). Also, a very large number said they have too much administrative work (100%), were confused regarding SD (98%), that ESD is irrelevant to their subject (90%), and their curriculum was too overcrowded to add ESD (84%). Seventy percent of those in Class 1 had a lack of principal support and suffered from class sizes that were too large (76%). Therefore, this class was named the "combination of individual and structural" barrier type. This type accounted for approximately as much as 43% of the teachers.

To provide a visible summary of the four-class model, represented as four different lines, the profile plot of the four-class model is given in Figure 2.

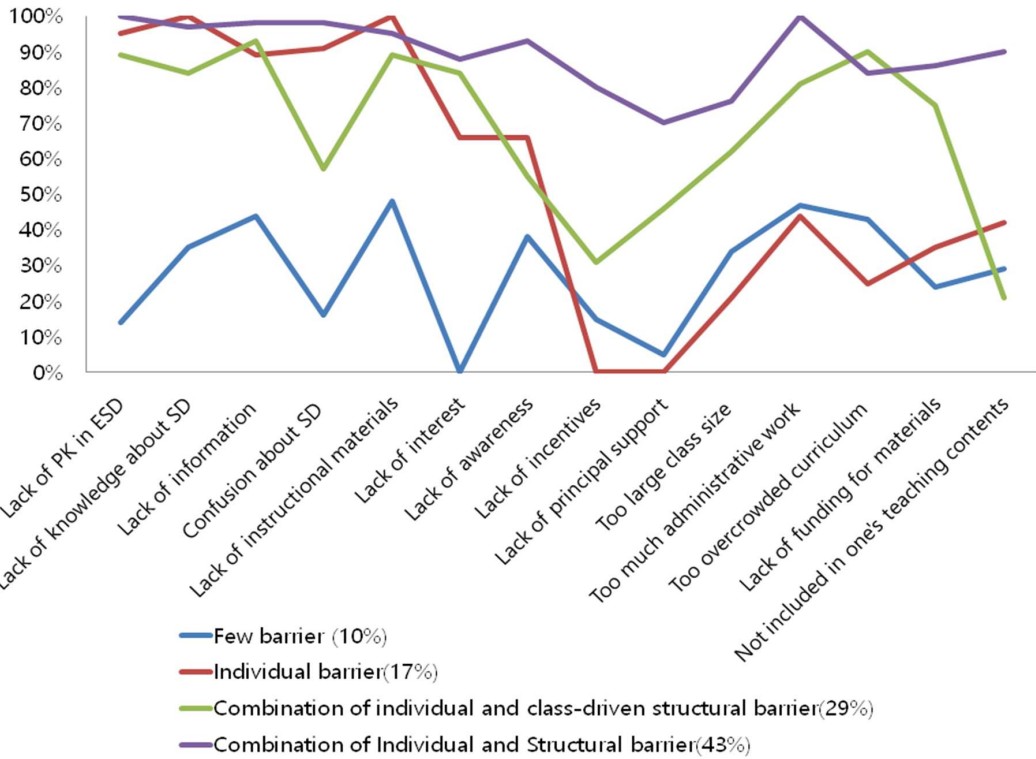

Note: The *y*-axis represents the item response probabilities for a teacher's class and the *x*-axis represents the categories for each variable.

**Figure 2.** Profile plot for all classes.

*4.3. Are Teachers' Experiences of Taking ESD Courses in Their Pre-Service Teacher Education and in Their In-Service Training on ESD Predictive of Membership in Perceived Barrier Types?*

To explore whether teachers' experiences of taking ESD courses in their pre-service teacher education and in their in-service training are predictive of an individual's membership in a class, I carried out multinomial logistic regression, where the dependent variable was latent rather than directly observed [38]. The combination of individual and structural barriers type (Class 4) was specified as the reference class for the multinomial logistic regression. Both covariates are dummy variables. "Experience (or participation)" was coded as "1", while "non-experience (or non-participation)" was coded as "0". The multinomial logistic regression results are displayed in Table 6.

**Table 6.** Parameter estimates and odds ratios for learning experiences on SD/ESD.

|  | Few Barrier | | Individual Barrier | | Combination of Individual and Class-Driven Structural Barrier | |
|---|---|---|---|---|---|---|
|  | Beta | Odds Ratio | Beta | Odds Ratio | Beta | Odds Ratio |
| Pre-service teacher education curriculum * | 1.51 | 4.54 | −2.76 | 0.06 | 0.74 | 2.09 |
| In-service training courses ** | 1.90 | 6.69 | −0.30 | 0.74 | −1.26 | 0.28 |

Note: Reference class is the Combination of Individual and Structural barriers type. * $p < 0.05$ ** $p < 0.001$.

As can be seen in Table 6, both the experience taken course in pre-service teacher education curriculum ($p < 0.05$) and participation in in-service training courses ($p < 0.001$) were significant predictors of latent class membership. Teachers who had taken ESD courses in pre-service teacher education were 94% less likely to be in the individual barrier type than the combination of individual and structural barrier type, and twice as likely to be in the combination of individual and class-driven structural barrier type than the combination of individual and structural barrier type. The most noticeable finding is that teachers who have taken ESD courses in pre-service teacher education were about five times more likely to belong in the few barrier types than the combination of individual and structural barrier type.

Teachers who have participated in in-service training on ESD were 26% less likely to be in the individual barrier type than the combination of individual and structural barrier type, and 70% less likely to be in the combination of individual and class-driven structural barrier type than the combination of individual and structural barrier type. The most striking finding is that teachers who have participated in in-service training on ESD were approximately seven times more likely to belong in the few barrier type than the combination of individual and structural barrier type.

## 5. Discussion

Several important findings that contribute to the understanding of teachers' perceived barriers to ESD resulted from this study.

First, the study results suggest that the frequency of the individual barrier component was very high.

Secondly, this study confirmed interactions between barriers and co-occurrence of teachers' perceived barriers using LCA. Furthermore, this survey identified four types of teachers exhibiting different profiles of perceived barriers to ESD implementation by LCA: the few barrier type (10%), the individual barrier type (17%), the combination of individual and class-driven structural barrier type (29%), and the combination of individual and structural barrier type (43%).

Thirdly, teachers' experiences of taking ESD course in their pre-service teacher education and in their in-service training on ESD were significant predictors of membership in perceived barrier types. Examination of the odds ratios for the significant predictor variables indicate which variables

accounted for probabilities belonging to each type compared reference group. Teachers who have taken courses on ESD in pre-service teacher education were about five times more likely to belong to the few barrier type than the combination of individual and structural barrier type. Also, teachers who have participated in in-service training on ESD were approximately seven times more likely to belong in the few barrier type than the combination of individual and structural barrier type. The findings suggest that pre-service teacher education and in-service training courses may be used as a critical vehicle to enhance ESD.

What do these results contribute to our understanding of teachers' perceived barriers to ESD and to facilitating ESD?

Firstly, it is an important step to identify barriers in exploring approaches for implementing ESD in the classroom. The key findings of the research elucidate the barriers for teachers in their ESD implementation. Since there is little research identifying barrier-type to ESD implementation among teachers, this research provides valuable information concerning types of the perceived barrier.

Secondly, this finding implied that when exposed to ESD in pre-service teacher education curriculum, teachers can develop informed understandings on SD. For this, it seems imperative that teacher education provide pre-service teachers with knowledge, skills, and attitudes needed for ESD. Furthermore, results of this investigation suggest that teacher educators may be stakeholders in addressing ESD in pre-service teacher education, and teacher educators in all fields have to be well-situated to address issues related to the integration of ESD in teacher education also.

Thirdly, participation in in-service training turned out to be one of the ways for improving the knowledge and competence of teachers. In-service training is traditional professional development that aims to improve an individual's skills, knowledge, expertise, and other characteristics as a teacher [39–41]. Recently, the sufficiency of these traditional professional development activities has been debated, and it has been recommended that effective professional development needs to go beyond traditional short-term teacher training courses facilitated by outside experts [42–45]. In spite of these arguments, this study proved that in-service training courses can improve teachers' knowledge and skills on ESD. Therefore, in-service training can be considered effective and worthy characteristics of professional development in the context of ESD. To combat the types of perceived barriers to ESD, the knowledge, skills, and attitudes needed should be taken into consideration during the design of in-service training courses.

Access to pre-existing materials can complement the class-driven structural barriers attributable to a lack of time and funding as well as a lack of materials. The Korea Foundation for the Advancement of Science and Creativity (KOFAC) and the Korean National Commission for UNESCO have been developing teaching materials on ESD in order to promote teachers to be well-equipped with the necessary information and teaching materials. To overcome a lack of incentives, policy-makers must ensure financial resources can be earmarked for activities related to ESD, and which offer benefits to schools to generate financial resources to invest in ESD projects. The principal is one of the most important influencers over teachers' decisions to implement ESD. Therefore, it is necessary to conduct principal training on ESD as a strategy so that the principal can share the vision and mission of ESD with the teachers.

## 6. Conclusion and Suggestions

The following conclusions can be made based on the findings of LCA. Firstly, there is a latent class structure that represents a heterogeneity in the barriers to ESD among Korean teachers, and 90% of teachers surveyed belonged to the type of perceived barriers based on individual components. Secondly, professional learning through both pre-service teacher education and in- service teacher training can reduce perceived barriers based on individual components.

The outcomes of the present research have implications for educational practice. Firstly, the successful implementation of ESD in secondary schools strongly depends on how competent teachers are in terms of ESD. Teachers surveyed perceived individual barriers related to competency deficiency

in terms of ESD. Lack of competency is not merely a matter of teachers, since teachers may be constrained by structural factors generating from the social system. This is something that is clearly related to both pre-service teacher education programs and to the structural support system including in-service training courses as well. Special attention must be paid to the structural factors along with the individual factors. Teacher education institutions have to keep up with their own responsibilities in this respect and try to prepare pre-service teachers in acquiring competencies for ESD. Policy-makers should develop in-service training courses on ESD, and support resources for expenses associated with in-service training so that teachers can participate in them.

While this research findings contribute to our understanding of teachers' perceived barriers to ESD and to facilitating ESD, some of the limitations must be considered.

First of all, there were limitations inherent in this research method. A quantitative survey research method was conducted in this research. The findings of this study may not be as in-depth as possible.

Secondly, non-probability convenience sampling was chosen as the sampling method in this study. Therefore, the results of this study might not be generalized to the whole of Korean secondary teachers. I suggest future studies attempt to replicate these results with probability sampling methods.

In addition to limitations, I would like to suggest the following for future studies. Firstly, teachers are more likely to implement ESD if there are many facilitators for implementing ESD. While identifying barriers is a crucial step in exploring approaches for ESD in secondary school, it is equally important to identify factors that may facilitate ESD in the school. This understanding helps teachers and policy-makers to emphasize facilitators, thereby increasing the likelihood of successful integration of SD in the existing subjects. I hope that researchers will identify facilitators in the future.

Secondly, in this study, I focused on the effects of both the experiences of taking ESD courses in their pre-service teacher education and in-service training on ESD on the types of perceived barriers. Both the form of the professional development (e.g., in-service training, professional learning community, workshops, and conferences) and the duration of professional development (e.g., the total number of hours that teachers participate in the activity) can be predictive of individuals' membership in each class. I suggest that researchers conduct future studies on the effects of both the form of the professional development and the duration of professional development so as to extend the stream of research in the literature on perceived barriers to ESD.

**Funding:** This research was funded by Daegu University.

**Acknowledgments:** This research was supported by Daegu University, Korea.

**Conflicts of Interest:** The author declares no conflict of interest.

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
