# Peer review of "Perceived Barriers to Implementing Education for Sustainable Development among Korean Teachers"

_sustainability, doi:10.3390/su11092532_

Round 1
Reviewer 1 Report
Suggestions/comments:
- Rewrite the Abstract, for example indicate objective, methods, conclusion and, if possible, some implication;
- Rewrite and reduce the long Keywords;
- Please, explain and justify with literature (Ln30-34);
- About Ln 35-40: for an international public its not obvious the author's argumentation. I suggest a better explanation;
- "identifying perceived barriers to ESD is, in my opinion, a good way to improve the possibilities of successful implementation." Justify this personal position;
- Its possible to find typos like "to this approach,";
- the "2. Literature Review" need to mobilize more references from the corpus of scientific knowledge of the research subject;
- I suggest a text transition between 2. and 3. chapters;
- I suggest some literature mobilization in Discussion.
Author Response
- Rewrite the Abstract, for example indicate objective, methods, conclusion and, if possible, some implication;
The manuscript has been modified to reflect comment (in the line 10-31).
We presented the relevant information in the line 252-258.
We supplemented the discussion and implication of this study in the line 339-391.
- Rewrite and reduce the long Keywords;
The manuscript has been modified to reflect comment (line 36-42).
- Please, explain and justify with literature (Ln30-34);
The manuscript has been modified to reflect comment (line 32-33).
- About Ln 35-40: for an international public its not obvious the author's argumentation. I suggest a better explanation;
The manuscript has been modified to reflect comment (line 43-45).
- "identifying perceived barriers to ESD is, in my opinion, a good way to improve the possibilities of successful implementation." Justify this personal position;
The manuscript has been modified to reflect comment (line 55-60).
- the "2. Literature Review" need to mobilize more references from the corpus of scientific knowledge of the research subject;
The manuscript has been modified to reflect comment
- I suggest a text transition between 2. and 3. chapters;
Chapter 2 provides a review of the literature concerning the literature concerning perceived barrier to ESD. Chapter 3 describes the methodology that my study uses including the sample, variables, statistics, and analysis. I think it's better for literature review to be situated ahead of methodology. I changed my research design to research methodology
- I suggest some literature mobilization in Discussion.
The manuscript has been modified to reflect comment
Thank you very much for your comments.
Reviewer 2 Report
This paper presents a research project dealing with analysis and identification of barriers in Barriers for
Implementing Education for Sustainable Development among Korean Teachers.
It seems like an interesting work, but I had trouble in reading this paper.
First sentence in the abstract is too long and it is not structured in accordance with a form and standards in writing an abstract. Also, I think this paper needs a substantial English langue. Editing.
However, the introduction is well written, even some part of the text I had trouble understanding. Some parts of the text in whole paper –I think it is because of bad choice of words, are unrelated.
Measurement instruments – whole chapter is very hard to understand. I am not sure that all important parts of methods, and processes are described (Cronbach?). It is (in some parts) confusing. Also, results chapter should be better organized. Discussion is well written and logical. The results presented in tables/figures are a very clear and transparent.
For these type of the study, just 20 references are used? That is odd.
General opinion: need English correction and editing, a major revision, because in this form is very confused.
Author Response
This paper presents a research project dealing with analysis and identification of barriers in Barriers for Implementing Education for Sustainable Development among Korean Teachers. It seems like an interesting work, but I had trouble in reading this paper.
First sentence in the abstract is too long and it is not structured in accordance with a form and standards in writing an abstract.
The manuscript has been modified to reflect comment
Also, I think this paper needs a substantial English langue. Editing.
However, the introduction is well written, even some part of the text I had trouble understanding. Some parts of the text in whole paper –I think it is because of bad choice of words, are unrelated.
I got a proofreading from a native speaker, and the manuscript has been modified to reflect it.
Measurement instruments – whole chapter is very hard to understand. I am not sure that all important parts of methods, and processes are described (Cronbach?). It is (in some parts) confusing. Also, results chapter should be better organized.
The manuscript has been modified to reflect comment
For these type of the study, just 20 references are used? That is odd.
The manuscript has been modified to reflect comment
General opinion: need English correction and editing, a major revision, because in this form is very confused.
I got a proofreading from a native speaker, and the manuscript has been modified to reflect it.
Thank you very much for your comments.

Round 2
Reviewer 1 Report
Some comments:
- Ln 44, Ln 74 seems to be formatted;
- clarify "1. Are there subgroups that adequately represent the heterogeneity in the perceived barriers to ESD among Korean teachers? If so, what are the types and their corresponding prevalence?";
- "5. Conclusion and Suggestion" are much better.
Author Response
1. Ln 44, Ln 74 seems to be formatted; The manuscript has been modified to reflect comment 2. clarify "1. Are there subgroups that adequately represent the heterogeneity in the perceived barriers to ESD among Korean teachers? If so, what are the types and their corresponding prevalence?"; The manuscript has been modified to reflect comment 3. "5. Conclusion and Suggestion" are much better. The manuscript has been modified to reflect comment My manuscript has improved considerably in the process of reflecting your comments. Thank you very much for your delicate review and comments.

Reviewer 2 Report
Good work in correction of the manuscript.
Author Response
My manuscript has improved considerably in the process of reflecting your comments.
Thank you very much for your delicate review and comments.